# Dynamic Asset Allocation with Expected Shortfall via Quantum Annealing

**DOI:** 10.3390/e25030541

**Published:** 2023-03-21

**Authors:** Hanjing Xu, Samudra Dasgupta, Alex Pothen, Arnab Banerjee

**Affiliations:** 1Department of Computer Science, Purdue University, West Lafayette, IN 47906, USA; xu675@purdue.edu (H.X.);; 2Department of Physics, Purdue University, West Lafayette, IN 47906, USA; 3Oak Ridge National Laboratory, Quantum Computing Institute, Oak Ridge, TN 37831, USA; 4Bredesen Center, University of Tennessee, Knoxville, TN 37996, USA

**Keywords:** portfolio optimization problem, Quadratic Unconstrained Binary Optimization (QUBO), quantum annealing, hybrid algorithm

## Abstract

Recent advances in quantum hardware offer new approaches to solve various optimization problems that can be computationally expensive when classical algorithms are employed. We propose a hybrid quantum-classical algorithm to solve a dynamic asset allocation problem where a target return and a target risk metric (expected shortfall) are specified. We propose an iterative algorithm that treats the target return as a constraint in a Markowitz portfolio optimization model, and dynamically adjusts the target return to satisfy the targeted expected shortfall. The Markowitz optimization is formulated as a Quadratic Unconstrained Binary Optimization (QUBO) problem. The use of the expected shortfall risk metric enables the modeling of extreme market events. We compare the results from D-Wave’s 2000Q and Advantage quantum annealers using real-world financial data. Both quantum annealers are able to generate portfolios with more than 80% of the return of the classical optimal solutions, while satisfying the expected shortfall. We observe that experiments on assets with higher correlations tend to perform better, which may help to design practical quantum applications in the near term.

## 1. Introduction

We describe a hybrid quantum-classical algorithm to solve a dynamic asset allocation problem where the targeted return and expected-shortfall (ES)-based risk appetite are specified. Since both the return as well as the shortfall are functions of the chosen asset allocation, we treat the return as a constraint in a modified Markowitz framework, and optimize the allocation strategy to meet the requirements of the expected shortfall using an iterative procedure that solves the Markowitz Optimization problem at each iteration. The latter optimization problem is solved by a Quadratic Unconstrained Binary Optimization (QUBO) formulation on a quantum annealer, while the iterative procedure to compute the shortfall is performed by a classical algorithm.

Quantum annealing offers a highly parallelized approach for solving optimization problems by using quantum tunneling from a manifold of high-energy solutions to the ground state. A common approach to embed the optimization problem into an Ising quantum annealer is to convert it to a QUBO problem [1,2,3,4]. Several examples have been explored so far in the literature, including the maximum clique [5], scheduling [6] and graph coloring problems [7], among others [8].

The portfolio optimization problem, introduced by Harry Markowitz [9] in 1952, investigates how investors could use the power of diversification to optimize portfolios by minimizing risk, and serves as a foundation for later models, such as the Black–Litterman model [10]. The original Markowitz Optimization problem used volatility as the measure of the risk. However, it is now known that volatility changes with time [11]; hence, treating it as a constant is risky and sub-optimal. Furthermore, it often fails to characterize the market during extreme events or “shocks”, for example, the 2008 mortgage crisis which led to an abrupt collapse of the market with the insolvency of Lehman Brothers. As a result, modern finance practitioners prefer to use a time-varying risk metric such as stochastic volatility, Value-at-Risk (VaR) or the expected shortfall. The latter is defined as the average loss that can be expected when the loss has already exceeded a specific threshold [12]. The advantages of expected shortfall over other risk measurements such as volatility or Value-at-Risk are discussed in [11].

It is NP-Hard to solve general quadratic optimization problems [13]. For convex quadratic optimization problems such as the portfolio optimization problem, however, there exist polynomial time algorithms that takes $O(n^{7/2}L)$ time [14], where *n* is the number of variables and *L* bounds the number of digits for each integer. Hence it is prohibitive to solve large-scale portfolio optimization problems exactly using classical methods due to the high time complexity. Hence as more versatile and scalable quantum computing devices-currently quantum annealers-enter the market, we explore solving the portfolio optimization problem on two such machines available today using QUBO formulations.

In Grant et al. [15], the authors have benchmarked the performance of a D-Wave 2000Q quantum annealer on solving the Markowitz Optimization Problem with a relatively small size of 20 logical variables and random data. Our study has the following novel contributions:We demonstrate how optimization problems with non-polynomial constraints such as the Expected Shortfall could be solved with a hybrid quantum-classical, iterative approach that requires no additional qubits. An alternative approach would encode such constraints directly into a QUBO by converting them first to a multilinear polynomial through Fourier analysis [16], and then to a quadratic polynomial using methods described in [17,18,19]. However, in this approach, in the worst-case the number of binary variables will grow exponentially due to non-trivial higher-order terms generated from the Fourier expansion, which severely limits the problem sizes that we can solve on the current generation of quantum hardware.To the best of our knowledge, quantum computing has not been employed prior to this study for solving Expected-Shortfall based dynamic asset-allocation problems [12]. Previous approaches (e.g., [15]) have employed the classical Mean-variance framework. However, static variance is no longer used in modern finance as it is well known that volatility fluctuates with time and hence it needs to be modeled in a statistical framework that captures non-stationarity. Moreover, industrial practitioners prefer tail-risk measures such as Value at Risk and Expected Shortfall (the latter is considered cutting edge in risk management) since true risk is associated with the fluctuations in the negative return, and is not symmetric with respect to positive and negative returns (i.e., no one minds a surprise positive return).Thirdly, this is one of the first papers that uses quantum computing for portfolio optimization using real financial data (using ETF and currency data) on a real quantum computer (i.e., not simulation) in an accurate industry setting. Previous approaches have used random data (e.g., [15]).

We further explored our algorithm’s performance on two generations of quantum annealers offered by D-Wave, with up to 115 logical variables. We provide experimental results on both the Advantage (Pegasus topology) and 2000Q (Chimera topology) D-Wave quantum annealers. The results are generally close to the optimal portfolios obtained by classical optimization methods, in terms of final returns and Sharpe ratios (return/standard deviation of the return in a time period).

The paper is structured as follows. Section 2 defines the expected shortfall based dynamic asset allocation problem and lays out a hybrid algorithm for solving it. Section 3 provides the technical background on D-Wave’s quantum annealer and maps the Mean-Variance Markowiz problem on it. Section 4 discusses the experimental results on both D-Wave 2000Q and Advantage systems. Section 5 states our conclusions and lists future research directions.

## 2. The Problem of Dynamic Asset Allocation

The problem of dynamic asset allocation is to allocate/invest an amount of money into *N* assets, while satisfying an expected return and keeping the risk below a given threshold. To make the problem more specific, we need to describe the input data and variables.

The historical return matrix *R* is obtained from Yahoo Finance [20,21,22,23,24,25] for the assets mentioned in Section 4.2 with *N* rows and $T_{\mathrm{total}}$ columns, where *N* is the number of assets and $T_{\mathrm{total}}$ is the number of days data are collected. We divide the return matrix *R* into periods of *T* days and index the data for each time period, for example, $R_{t}$ represents the return data from *t*-th time period.The vector of asset means $\mu_{t}$ is computed from $R_{t}$.The co-variance matrix $C_{t}$ calculated from the matrix $R_{t}$ as
(1)$$C_{t,i,j}=\frac{\left(e_{i}^{T}R_{t}-\mu_{t,i}^{T}1\right)\;\cdot \;\left(e_{j}^{T}R_{t}-\mu_{t,i}^{T}1\right)}{T-1},$$
where $e_{j}$ is the column vector of all zeros except with a one at the *j*-th position.

Asset allocation is especially interesting to financial practitioners during a time period with unpredictable market turbulence with goal of minimizing risk while achieving a target return. The risk is upper bounded by a consumer-driven risk appetite. A data sheet of the assets’ daily returns (profit one can earn if buying an asset the previous day and selling the next) for the previous three months is available. The risk threshold can be set using observed market metrics from a volatile time period, for example, the 2008 market crash. The algorithm uses the assets’ historical return data to estimate the trends of the assets’ performance and their correlations. Options for risk measurements include:Volatility: the standard deviation of the portfolio return.Value-at-Risk at level α: the smallest number *y* such that the probability that a portfolio does not lose more than y% of total budget is at least 1−α.Expected Shortfall at level α: the expected return from the worst α% cases. It is defined as follows:
(2)$${\mathrm{ES}}_{\alpha }\left(w_{t},R_{t}\right)=\mathrm{mean}\left(\mathrm{lowest}\;\alpha \%\;\mathrm{from}\;w_{t}^{T}R_{t}\right).$$

We focus on the expected shortfall as our risk measurement for the rest of the paper as it is the modern approach preferred by practitioners (as mentioned earlier in Section 1). The problem can be expressed as follows where the weight vector *w* indicates what fraction of the budget is invested in each asset:

(P1) Minimize the expected shortfall ${\mathrm{ES}}_{\alpha }\left(w_{t},R_{t}\right)$ under the constraints that the expected return is satisfied, the variance of the portfolio is small, and all assets are invested.

It is possible to write the expected shortfall based portfolio optimization as a linear program [26], but it requires adding N+1 variables and 2N constraints where *N* is the number of assets. Since the expected shortfall cannot be expressed by a quadratic formulation natively, we opt to use it as a convergence criterion instead of including it in the optimization problem directly. To justify this approach, assuming that the assets’ historical returns follow a Gaussian distribution, we can approximate the expected shortfall of a given portfolio *P* by:(3)$${\mathrm{ES}}_{\alpha }\left(P\right)=\mu +\sigma \frac{\phi (\Phi^{-1}\left(\alpha \right))}{1-\alpha },$$
where μ is the expected return, σ is the volatility of the portfolio, and ϕ(x) and Φ(x) are the Gaussian probability distribution and cumulative distribution functions, respectively [27]. The expected shortfall has positive correlation with the volatility and in turn, the variance of the portfolio ([11,28]).

Hence we propose a bilevel optimization approach descried in Figure 1 to solve Problem (P1).

Given a balance sheet of the assets’ returns in the history, we create multiple time periods each with *T* days. After picking target return $p_{t}$ for one of the time periods *t*, we choose a reference asset that is representative of the portfolio, and set a target expected shortfall for that asset computed from its volatility in the year 2008, its volatility in the time period *t*, and its shortfall in 2008. (The precise expression is included in Algorithm 1). Then we use the Markowitz Optimization problem [9] to allocate assets within the portfolio in order to minimize volatility with the constraint that the target return is met. Next we compute the expected shortfall from the current allocation of assets. If the target expected shortfall is not met by the current allocation, then we adjust the target return value and iteratively solve the Markowitz Optimization problem. We terminate when the target expected shortfall is met, or the target return cannot be met as the maximum return of all assets is smaller than the target return.

Now we describe the Markowitz portfolio optimization procedure. Its QUBO formulation will be provided in the next section. The Markowitz Optimization problem can be expressed by the quadratic optimization problem
(4)$$\begin{array}{cc} \min_{w}\; & w_{t}^{T}C_{t}w_{t}\\ s.t.\; & \mu_{t}^{T}w_{t}=p_{t},\\ \; & \sum_{i}w_{t,i}=1,w_{t,i}\geq 0\;\forall i. \end{array}$$
where $p_{t}$ is the target portfolio return during *t*-th time period. The constraint $\mu_{t}^{T}w_{t}=p_{t}$ ensures that the target return is met, $\sum w_{t,i}=1$ indicates we want to invest all of the resources, and $w_{t,i}\geq 0$ means short selling is not allowed. With all the constraints satisfied, we minimize $w_{t}^{T}C_{t}w_{t}$, that is, the variance of the portfolio at t−th time period. However, with our bilevel optimization, we treat the constraints as soft constraints, that is, small violations of their values are permitted. The optimizer can return portfolios with small variance even when the expected return falls short of the target; if the sum of weight is not equal to 1, we can scale the weights of the assets to sum to 1.
**Algorithm 1:** Expected Shortfall based Dynamic Asset Allocation during *t*.
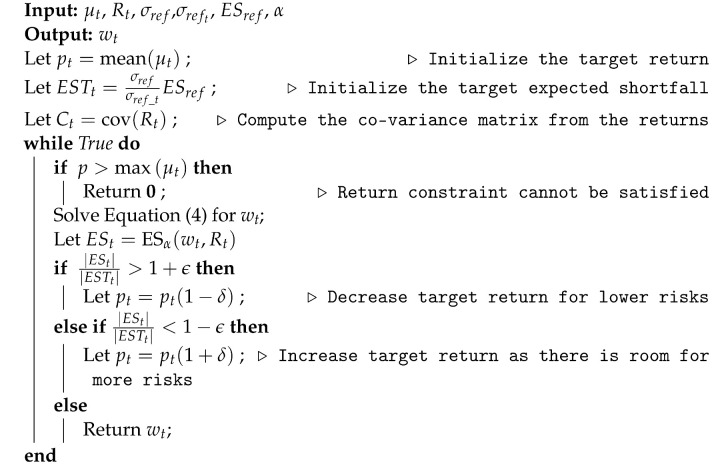


Algorithm 1 provides the pseduo-code for the algorithm for expected shortfall based asset allocation. Here $\sigma_{ref}$ is the volatility of a reference asset’s returns during the market crash in 2008; the reference asset is chosen from among the assets to be representative of the market trend, for example, SPDR S&P 500 ETF Trust (SPY). The variable $\sigma_{ref_{t}}$ is the volatility of the reference asset’s returns during the time window *t*; $ES_{ref}$ is reference asset’s expected shortfall during the market crash; $EST_{t}$ is the target expected shortfall at time window *t*; α is the risk level parameter; $ES_{t}$ is the expected shortfall for the computed portfolio during the optimization process at time window *t*; ϵ is the error tolerance parameter; and δ is the momentum parameter that is adjusted dynamically. Figure 2 shows the ratio between the variance and expected shortfall in different iterations of Algorithm 1 from ETFs consisting of 6 assets whose returns were obtained from December 2019 to May 2020. The monotonic one-to-one tracking justifies why optimization problems with expected shortfall constraint can be solved iteratively using the Markowitz Mean-Variance framework.

## 3. A Hybrid Quantum Classical Algorithm

### 3.1. Algorithm Overview

We will use a hybrid quantum classical algorithm to solve the quadratic optimization problem given in Equation (Equation 4) with a quantum annealer backend.

Quantum annealing (QA) [29,30,31] is the quantum analog of the classical annealing where the disorder is introduced quantum mechanically instead of thermally via applying the Pauli matrix *x* on every qubit as in Equation (Equation 5).
(5)$$H_{I}=-\sum_{i=1}^{N}\sigma_{i}^{x},$$
This Hamiltonian does not commute with the problem Hamiltonian in Equation (Equation 6)
(6)$$H_{P}=-\sum_{i}h_{i}\sigma_{i}^{z}-\sum_{i<j}J_{ij}\sigma_{i}^{z}\sigma_{j}^{z},$$
where $\sigma_{i}^{z}$ is the Pauli matrix *z* acting on qubit *i*, $h_{i}$ is the magnetic field on qubit *i* and $J_{ij}$ defines the coupling strength between qubits *i* and *j* [32]. The spin configurations of the ground states of Equation (Equation 6) also minimize the Ising model problem:(7)$$\min_{s}\;E\left(s\right)=-\sum_{i}h_{i}s_{i}-\sum_{i,j}J_{ij}s_{i}s_{j},\;s_{i}\in \left\{-1,1\right\},$$
where $s_{i}$ is the spin, *h* is the external longitudinal magnetic field strength vector and the matrix *J* represents the coupler interactions. Moreover, the general two-dimensional Ising problem within a magnetic field is NP-hard [33]. And in the case of spin-glass three-dimensional Ising model with lattice size of N=lmn, the complexity is $O(2^{mn})$ [34], which is NP-Hard as well.

During the QA process, combining both Hamiltonians in Equations (Equation 5) and (Equation 6), at time *t* the system evolves under the following Hamiltonian:(8)$$H\left(t\right)=A\left(\frac{t}{T}\right)H_{I}+B\left(\frac{t}{T}\right)H_{P}.$$
Here *T* is the total annealing time and the system is initialized to the ground state of $H_{I}$, which is a superposition of all qubits in the *z* basis. Functions $A(\frac{t}{T})$ and $B(\frac{t}{T})$ describe the change of influences from disorder and problem Hamiltonians on the system. $H_{I}$ dominates $H_{P}$ initially and slowly (adiabatically) changes to the opposite while the influence of $H_{I}$ vanishes at the end of the annealing process, thus removing disorder from the system. The system will then settle into one of the low energy states.

Due to unavoidable experimental compromises [35], QA serves as an intermediate step towards universal adiabatic quantum computation (AQC) [36,37] as the system evolves under a time-dependent Hamiltonian
(9)$$H=\left[1-s\left(t\right)\right]H_{I}+s\left(t\right)H_{P},$$
where s(t) changes from 0 to 1. When conditions on internal energy gap and time scales are met [38], the system will remain in its ground state at all times, which is different from QA.

A Quadratic Unconstrained Binary Optimization (QUBO) problem of the form
(10)$$\min_{x}\;Q\left(x\right)=\sum_{i}h_{i}x_{i}+\sum_{i,j}J_{ij}x_{i}x_{j},\;x_{i}\in \left\{0,1\right\}$$
aims to minimize a mathematical function with linear and quadratic terms; here any combination of $x_{i}\in \left\{0,1\right\},\forall i$ is feasible. It can be converted to the Ising model shown in Equation (Equation 7) by a one-to-one mapping of the variables: $x_{i}=\frac{1+s_{i}}{2}$. We will use the QUBO formulation for the rest of the paper but note that quantum annealers from D-Wave require the QUBO problems to be transformed into Ising models before execution.

Consider a standard binary optimization problem with a linear or quadratic objective function f(x) and linear constraints Ax=b, where $\;A\in R^{m\times n}$, and $b\in R^{m\times 1}$.
(11)$$\begin{array}{cc} \; & \min_{x}\;f\left(x\right)\\ \; & s.t.\;Ax=b,\\ \; & \mathrm{where}\;x\in {\left\{0,1\right\}}^{n\times 1}. \end{array}$$
We can rewrite it as a QUBO
(12)$$Q\left(s\right)=f\left(x\right)+\lambda {\left(Ax-b\right)}^{T}\left(Ax-b\right)$$
to be minimized by quantum annealers with a large enough $\lambda \in R^{+}$ to guarantee that the constraint is satisfied in the optimal solutions.

We will now discuss how to convert the Markowitz Optimization problem with continuous variables in Equation (Equation 4) to a QUBO problem.

First we write Equation (Equation 4) as an unconstrained optimization optimization problem with penalty coefficients $\lambda_{1}$ and $\lambda_{2}$ (the subscripts *t* are dropped for better readability):(13)$$Q=\left(\sum_{i}^{n}\sum_{j}^{n}C_{i,j}w_{i}w_{j}\right)+\lambda_{1}{\left(\sum_{i}^{n}\mu_{i}w_{i}-p\right)}^{2}+\lambda_{2}{\left(\sum_{i}^{n}w_{i}-1\right)}^{2},$$
where $\lambda_{1}$ and $\lambda_{2}$ scale the constraint penalties. Minimizing Equation (Equation 13) is equivalent to
(14)$$\begin{array}{c} \min Q=\left(\sum_{i}^{n}\sum_{j}^{n}C_{i,j}w_{i}w_{j}\right)+\lambda_{1}\left[{\left(\sum_{i}^{n}\mu_{i}w_{i}\right)}^{2}-2p\sum_{i}^{n}\mu_{i}w_{i}\right]\;\\ \;+\lambda_{2}\left[{\left(\sum_{i}^{n}w_{i}\right)}^{2}-2\sum_{i}^{n}w_{i}\right], \end{array}$$
after expanding the squared terms and eliminating the constants. When the constraints are satisfied exactly, we have
(15)$$\lambda_{1}\left[{\left(\sum_{i}^{n}\mu_{i}w_{i}\right)}^{2}-2p\sum_{i}^{n}\mu_{i}w_{i}\right]=-\lambda_{1}p^{2},$$
and
(16)$$\lambda_{2}\left[{\left(\sum_{i}^{n}w_{i}\right)}^{2}-2\sum_{i}^{n}w_{i}\right]=-\lambda_{2}.$$

We use *k* binary variables $x_{i,1}\ldots x_{i,k}\in \left\{0,1\right\}$ to approximate each continuous variable $w_{i}$ in Equation (Equation 4) with a finite geometric series
(17)$$w_{i}=\sum_{a=1}^{k}2^{-a}x_{i,a}.$$

The larger *k* is, the more precision $w_{i}$ has. However, larger *k* also widens the differences between the coupler strengths—*J* terms in Equation (Equation 10). Although the coupler strengths for D-Wave annealers can be set at any double-precision floating point number between −1 and 1, precision errors may pose a challenge due to integrated control errors (ICE) [39]. In our experiments, we set k=5 which empirically gives us the best approximations to the optimal solutions for Equation (Equation 4). For larger *k* we risk the errors dominating the coupler coefficients, rendering those additional qubits unreliable. We set $\lambda_{1}$ to $p^{-2}$ and $\lambda_{2}$ to 1 to bound the penalty terms in Equation (Equation 14) to −2. Additionally, we scale the objective by a factor of $\lambda_{3}$ to around 1 such that penalty terms in the optimal solutions to Equation (Equation 14) remain relatively small while not overwhelming the objective. If penalties dominate the objective, it may introduce numerous local minima to the energy landscape and the optimizer will suffer from barren plateaus. Alternatively, if the objective dominates penalties, constraints will be violated significantly. The soft constraints enable us to obtain better portfolios as presented in Section 4.4.

Substituting Equation (Equation 17) in to Equation (Equation 14), we have the final binary optimization formalism
(18)$$\begin{array}{c} f\left(x\right)=\left[{\left(\sum_{i}^{n}\sum_{a}^{k}\mu_{i}2^{-a}x_{i,a}\right)}^{2}-2p\sum_{i}^{n}\sum_{a}^{k}\mu_{i}2^{-a}x_{i,a}\right]\;\\ +p^{-2}\left[{\left(\sum_{i}^{n}\sum_{a}^{k}2^{-a}x_{i,a}\right)}^{2}-2\sum_{i}^{n}\sum_{a}^{k}2^{-a}x_{i,a}\right]\\ \;+\lambda_{3}\sum_{i}^{n}\sum_{j}^{n}\sum_{a}^{k}\sum_{b}^{k}C_{i,j}2^{-a-b}x_{i,a}x_{j,b}, \end{array}$$
which is quantum-annealable as it only has linear and quadratic interactions.

### 3.2. Previous Work

Rosenberg et al. [40] solve the multi-period portfolio optimization problem using D-Wave’s quantum annealer:(19)$$\begin{array}{cc} \max_{w}\; & \sum_{t=1}^{T}(\mu_{t}^{T}w_{t}-\frac{\gamma }{2}w_{t}^{T}\Sigma_{t}w_{t}-\Delta w_{t}^{T}\Lambda_{t}\Delta w_{t}\\ \; & +\Delta w_{t}^{T}\Lambda_{t}^{{}^{\prime }}\Delta w_{t})\\ s.t.\; & \sum_{n=1}^{N}w_{nt}=K,\forall t,\;w_{nt}\leq K^{\prime },\forall t,\forall n. \end{array}$$
Here *T* is the number of time steps, and *N* is the number of assets. At each time step *t*, $\mu_{t}$ represents the forecast returns, $w_{t}$ are holdings for each asset, $\Sigma_{t}$ is the forecast covariance matrix, $\Lambda_{t}$ and $\Lambda_{t}^{{}^{\prime }}$ are coefficients for transaction costs related to temporary and permanent market impacts, respectively, which penalize changes in the holdings if the corresponding terms are positive. Additionally, γ is the risk aversion factor.

Equation (Equation 19) seeks to maximize returns considering constraints on asset size. Specifically, the sum of asset holdings is constrained by *K* and the maximum allowed holdings of each asset is $K^{\prime }$. For small problems ranging from 12 to 584 variables, D-Wave’s 512 and 1152-qubit systems are able to find optimal solutions with high probability.

Venturelli and Kondratyev [41] focus on the following QUBO problem where the task is to select *M* assets from a pool of *N* assets:(20)$$\min_{q}\;\sum_{i=1}^{N}a_{i}q_{i}+\sum_{i=1}^{N}\sum_{j=i+1}^{N}b_{ij}q_{i}q_{j}+P\left(M-\sum_{i=1}^{N}q_{i}\right).$$
The variable $q_{i}$ is 1 if asset *i* is selected and 0 otherwise. The coefficient $a_{i}$ indicates the attractiveness of the *i*-th asset and $b_{ij}$ is the pairwise diversification penalties (positive) or rewards (negative). The penalty coefficient *P* scales the constraint on the number of selected assets to make sure it is satisfied in the optimal solution. The authors have explored the benefits of reverse annealing on D-Wave systems, and report one to three orders of magnitude speed-up in time-to-solution with reverse annealing.

The problem considered by Phillipson and Bhatia [42] is similar to the Markowitz Optimization problem but with binary variables indicating asset selections instead of real weights. The authors report comparable results from D-Wave’s hybrid solver to other state of the art classical algorithms and solvers including simulated annealing [43,44], genetic algorithm [45,46], linear optimization problems [47] and local search [48].

Grant et al. [15] benchmark the Markowitz Optimization problem on a D-Wave 2000Q processor with real weight variables and price data generated uniformly at random, and explore how embeddings, spin reversal and reverse annealing affect the success probability. Hegade et al. [49] solve the same problem with added counterdiabatic terms on circuit-based quantum computers and see improvements on success probabilities using digitized-adiabatic quantum computing (DAdQC) and Quantum Approximation Optimization Algorithm (QAOA) [49].

We extend the general QUBO formulation in [15] to solve the asset allocation problem, with the expected shortfall as the risk metric, using the Markowitz Optimzaiton problem as a subroutine at each iterative step of the algorithm. We use our algorithms on real-world ETF and currency data. Additionally, we present the results on the newly-available Advantage processor and experiment on problems with up to 115 logical variables, up from 20 in [15].

## 4. Experimental Setup and Results

### 4.1. D-Wave Quantum Annealer

We start by discussing the latest quantum annealing technologies offered by D-Wave as the solvability of the problem is dependent on the architecture. D-Wave quantum annealers are specifically designed to solve Ising problems natively. Currently two types of quantum annealers are offered by D-Wave: 2000Q processor with Chimera topology and the Advantage processor with Pegasus topology. The latter was made publicly available in 2020 and it has more qubits (5760 vs. 2048) and better connectivity than the former. The qubits in the Chimera topology have 5 couplers per qubit while in the Pegasus topology they have 15 couplers per qubit [50]. It is not always possible to formulate an optimization problem to match the Chimera or Pegasus topologies exactly. Therefore minor embeddings are necessary to map the problems to D-Wave processors. Such embeddings usually require the users to map multiple physical qubits to one logical variable with constraints such that every qubit on the ‘chain’ behaves the same, which significantly reduces the total size of the problems that can be solved on the quantum annealers.

Furthermore, it is advisable to have uniform chain lengths (number of qubits representing a single variable) for more predictable chain dynamics during the anneal [51]. Algorithms in [52] detail such procedures for fully-connected graphs which is the underlying logical graph for the portfolio optimization problem. A full-yield 2000Q processor can map up to 64 logical variables and an Advantage processor can map around 180 logical variables. A comparison between the embedding of the two topologies is shown in Figure 3. In our experiments, we use the *find_clique_embedding* function from *dwave-system* to map fully-connected graphs to either the Chimera or the Pegasus topology.

### 4.2. Test Input and Annealer Parameters

We pick the top-six ETFs by trading volumes, EEM, QQQ, SPY, SLV, SQQQ and XLF, and six major currencies’ USD exchange rates, AUD, EUR, GBP, CNY, INR and JPY, for most of the tests below. The reference assets for ETF and currency tests are SPY and EUR, respectively. For the tests in Section 4.5 we use 12 and 23 assets respectively and pick the top ETFs by trading volumes again. We choose the parameter α in the definition of expected shortfall to be 5%.

We can control a range of annealer parameters that may impact the solution quality in various degrees. Specifically, we set the number of spin reversal transforms [53] to 100 and readout thermalization to 100 μs as suggested in [54,55]. The spin reversal transform flips the signs of 100 variables and coefficients of the Ising model, which leaves the ground state invariant. The goal is to average out the system errors thus improving the quality of the solutions [53]. 100 μs readout thermalization allows the system enough time to cool back to the base temperature after each anneal. We set the annealing time at 1 μs as longer annealing time sees no statistically significant improvements to the solutions similarly reported in [15]. Results from 2000Q and Advantage processors are both included in the following sections. Additionally, we report the results from D-Wave’s post processing utility on 2000Q processors, which decomposes the underlying graph induced by the QUBO into several low tree-width subgraphs [56], and then solves them exactly using belief propagation on junction trees [57].

We sample all QUBOs 30,000 times with both D-Wave backends and report the samples with the lowest objective value from Equation (Equation 17) each time. Figure 4 shows an example distribution of the samples.

### 4.3. Embedding Comparison on D-Wave Annealers

As discussed in Section 4.1, D-Wave quantum annealers require the problems to be minor embedded to the Chimera or Pegasus topology. For small problems this means there may be multiple valid embeddings and in this section we will measure how different embeddings can make an impact on the solution quality.

We compute four different embeddings that use different sets of physical qubits from both 2000Q and Advantage processors. Otherwise, the embedding graphs are the same, and hence they use the same number of qubits and chain lengths. We sample the same QUBO—first iteration of Algorithm 1 on the ETFs from December 19 to May 20—10 times with 10,000 samples each. We then pick the best solutions in terms of QUBO objective value from all 10 sample sets for each embedding and obtain their average and minimum values. Table 1 and Table 2 report the results as ratios against the best objective values computed by simulated annealing for better readability. Since the objective values are negative, we compute ratios of the magnitudes instead.

We can see from Table 1 and Table 2 that the impact that different embeddings make is statistically insignificant. However, it is clear that the Advantage processors have higher ratios than the 2000Q processors, which we will address next.

### 4.4. Annealing Results Comparison

We benchmark our algorithm on both simulated and quantum annealers using, as the baseline algorithm, a classical optimization solver, namely, cvxpy [58]. We create five ETF test datasets and four currency test datasets from 100 days of return data with different starting dates from 2010 to 2020.

The results are normalized against the optimal classical solution. The quantum algorithm fails to converge for the first two currency tests on the 2000Q processor, and the corresponding bars are missing in Figure 5 and Figure 6.

In Figure 5 and Figure 6, we used k=5 binary variables to represent each asset weight. The simulated annealing results follow the optimal solutions closely in most tests. We note that in tests 2 and 5 from the ETF tests and tests 1, 2 and 3 from Currency tests, simulated annealing, and in some cases, quantum annealing produce portfolios of higher returns than those of the exact classical quadratic optimization problem solver. This is due to how Markowitz Optimization problems are formulated as QUBOs with discretized variables in Equation (Equation 18), which changes the optimization problem slightly, and also the optimal asset allocations. In test 4 from Currency tests, quantum annealers are able to find a portfolio with higher returns than simulated annealing as it returns a portfolio with a slightly increased risk that is still acceptable, but higher returns. This is not optimal in terms of QUBO objective values as the constraint penalty is now higher, yet the solution is still feasible. We also observe that the currency tests generally perform better than ETF tests on both quantum annealer backends. Figure 7 shows how quantum annealers perform with respect to the average of absolute correlation coefficients over all pairs of assets in each test. Higher correlation coefficients seem to lead to higher returns.

Although we acknowledge there may be other factors contributing to our observations that currency tests do better than ETF tests on for quantum annealers, Figure 7 implies that more correlated assets tend to perform better. Detailed analysis on which attributes of the assets have an impact on the quantum annealing performance and how much the impacts are requires more research in the future. Ref. [59] used machine learning models such as decision tree and regression to predict the accuracy of D-Wave’s quantum annealer on maximum clique problems.

### 4.5. State-of-the-Art on D-Wave Annealers

The embeddings of the six asset tests on both 2000Q and Advantage processors leave plenty of unused qubits. D-Wave’s clique embedding algorithm [52] suggests that we can embed fully connected graphs with 64 and 180 vertices to full-yield 2000Q and Advantage processors, respectively. Due to the defective qubits and connectors in the currently available Advantage processor, experimentally, we can embed only up to 119 qubits. This means we can solve portfolio optimization problems with 12 and 23 assets natively on 2000Q and Advantage processors, respectively.

On the 12 asset test shown in Figure 8, the 2000Q processor struggles to find the ground state as its embedding chain length reaches 16, while the Advantage processor provides results close to the simulated annealing and post-processed results. However, neither quantum annealer converges. Table 3 records the QUBO objective values of the last five iterations for the Advantage processor in this test. Although the objective values hardly differ, the solution quality is seemly more sensitive to changes in the QUBO objective value for larger problems. A 0.1% change in the objective value leads to 30% difference in the portfolio variance. One potential reason is that larger problems have more assets that are less correlated, and as shown in Figure 7, smaller correlation coefficients generally equate to worse performance on quantum annealers. In this case, either the quantum annealers need to be more accurate to find the ground state, or our QUBO setup needs to be modified to account for higher asset counts.

For even larger problems of 23 assets, with the embedding chain lengths going up to 17, the Advantage processor fails to find the ground state by a large margin, as shown in Figure 9. Even though we can physically map a problem of this size, the results reflect the limitations of current-generation quantum annealers.

## 5. Discussion

As newer quantum devices are released every year, it is important to design and benchmark algorithms across generations. As companies and researchers race to build the first quantum computer that can demonstrate quantum advantage on practical problems, different classes of quantum devices have emerged: general purpose quantum computers from IBM, Google, Honeywell, and others; the specialized quantum Ising machine from D-Wave; and quantum-inspired digital annealer from Fujitsu. These devices have different types of constraints due to different noise profiles, qubit connectivity, and/or implementable Hamiltonians, and none are perhaps at the scale and reliability needed to solve real-world problems at the edge of classical capability. Therefore, hybrid algorithms are needed to incorporate these quantum computers on practical problems with reasonable size.

In this paper, we have shown that it is not only possible to introduce such hybrid algorithm schemes that compute the optimal portfolios based on expected shortfall, but also highlighted where it is possible to reach working accuracy. We used a quantum annealer to solve an asset allocation problem based on expected shortfall, employing a QUBO formulation of the Markowitz Optimization problem and interlacing it with a layer of classical decision-making. Here, we iteratively adjusted our problem Hamiltonian based on its feedback until the portfolio was within the desired risk threshold. The fact that both D-Wave 2000Q and Advantage quantum annealers performed reasonably well on the six-asset tests with portfolios’ Sharpe ratios to above 80% of SA values is promising. Additionally encouraging is that the newer and more scalable Advantage processor achieved much better QUBO objective values on problems with 12 assets. Finally, we observed that both quantum annealers tended to obtain portfolios with higher returns on more correlated assets (Figure 7), which we believe should attract future research as it may help guide the application of quantum annealing on real-world applications in the near term.

Although the quantum annealers fell short on tests with more assets, we can remain optimistic about new hardware with more qubits, better connectivity, and lower noise in the near future. We also acknowledge the need to design algorithms that can scale with these new hardware, as we saw that the portfolio quality became increasingly sensitive to the QUBO objective values as we introduced more assets—results with 99.9% objective values of the optimal led to 30% more variance. Additionally, advances in gate-model quantum computers and combinatorial optimization algorithms [60,61] will provide other avenues for solving these problems. For example, it could be instructive to explore and compare to novel approaches, such as counterdiabatic techniques recently proposed for similar problems, but for gate-based systems [49].

Future research includes identifying subsets of problems that can be solved better on quantum devices, as we have discussed in Section 4. It is also important to find an efficient way to implement inequality constraints, as adding slack variables may not be the best choice in the QUBO. We also note that on specific test cases, the QUBO reformulation enables both simulated annealer and quantum annealers to find better portfolios than a classical convex optimizer cvxpy, by treating the constraints as soft. Other optimization problems might also benefit from QUBOs with soft constraints.

## Figures and Tables

**Figure 1 entropy-25-00541-f001:**
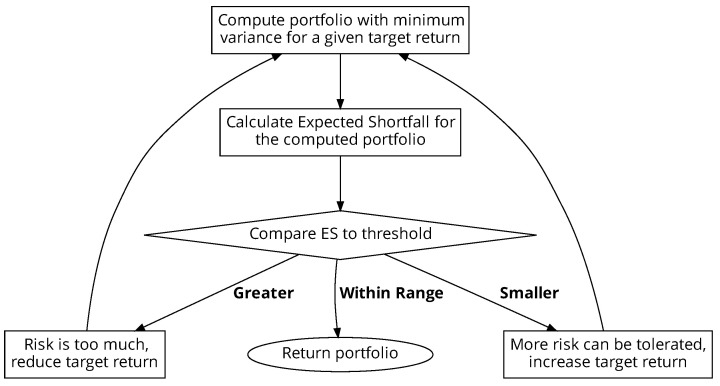
A flowchart for the proposed algorithm for computing optimal portfolio with a threshold on the expected shortfall.

**Figure 2 entropy-25-00541-f002:**
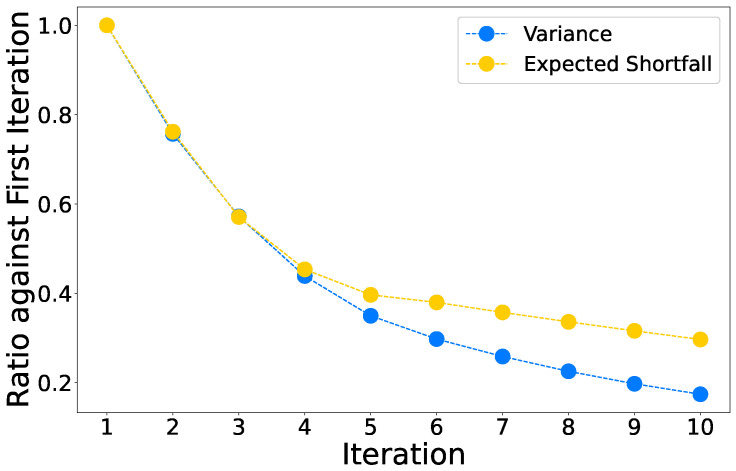
The *y*-axis tracks the ratio between the variance and expected shortfall with α=5 in later iterations against their respective values in the first iteration of Algorithm 1 running on 6 ETF assets. The expected shortfall decreases at a different rate from the variance but each iteration in the algorithm is guaranteed to make progress towards the target expected shortfall, which ensures convergence.

**Figure 3 entropy-25-00541-f003:**
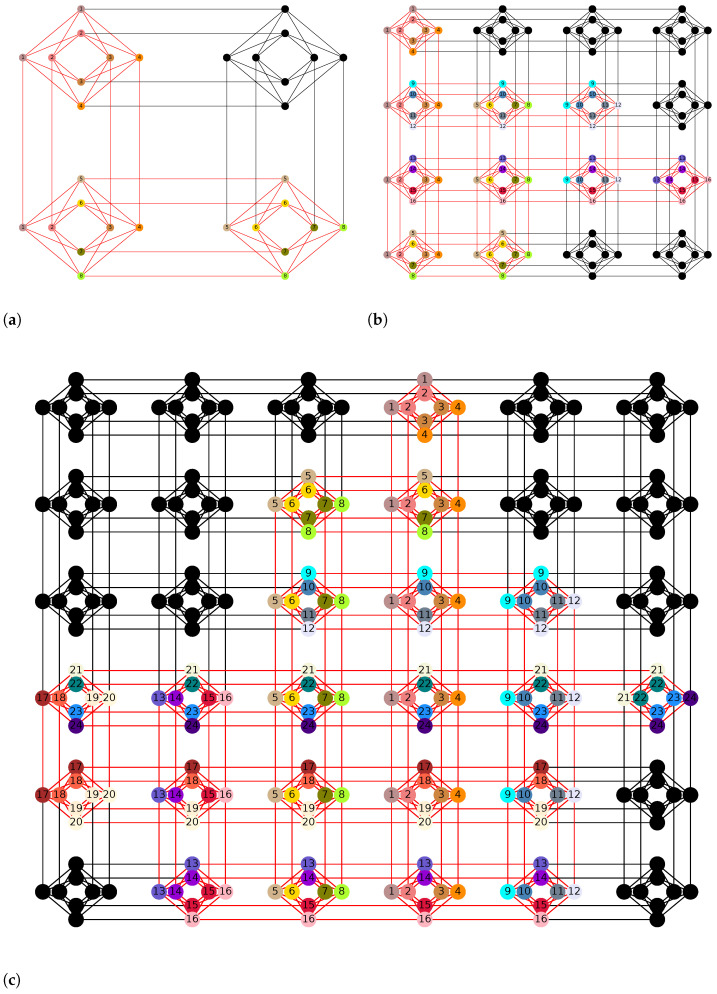
A comparison between minor embeddings on Chimera (2000Q) and Pegasus (Advantage) lattices of D-Wave processors for cliques (fully connected graphs) of different sizes. The vertices with the same color or label represent the same logical variable in Equation (Equation 7) and the chain length is defined as the number of qubits used to represent one logical variable. Each Chimera cell is a 4 by 4 complete bipartite graph ($K_{4,4}$) with 4 additional edges connecting neighboring cells. Each Pegasus cell has 24 qubits which include three $K_{4,4}$ graphs as in the Chimera cell and the cells are connected with each other using $K_{2,4}$ edges. To minor embed cliques of 8 vertices (K=8), the chain length on the Chimera lattice is 3 while on the Pegasus lattice it is 2. With K=16, the chain lengths are 5 and 2–3, respectively, and with K=24, they are 7 and 3–4, respectively. This shows that Pegasus processor scales better for larger clique problems, which may lead to better performance. (**a**) Embedding $K_{8,8}$ on Chimera topology; (**b**) Embedding $K_{16,16}$ on Chimera topology; (**c**) Embedding $K_{24,24}$ on Chimera topology; (**d**) Embedding $K_{8,8}$ on Pegasus topology; (**e**) Embedding $K_{16,16}$ on Pegasus topology; (**f**) Embedding $K_{24,24}$ on Pegasus topology.

**Figure 4 entropy-25-00541-f004:**
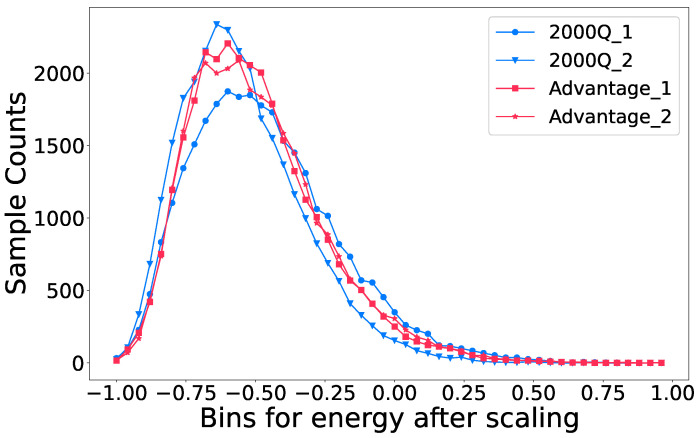
The distribution of the samples for 4 different QUBOs with both D-Wave backends. Each QUBO is sampled 30,000 times and the objective values of the samples is scaled to be between (−1, 1). We divide the objective value range into 50 equally-spaced bins and count the number of samples in each bin. All four samples exhibit the Poisson distribution, and thus we only report the samples with the lowest objective value for the experiments in this section since they can be reproduced reliably.

**Figure 5 entropy-25-00541-f005:**
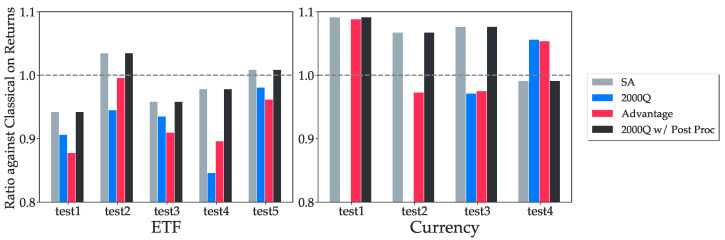
The comparison of the final returns between all four backends. A higher ratio means the backend can return portfolios with higher returns. Each test uses 100 days of return data with different starting dates from 2010 to 2020. The results from 2000Q with post processing yields identical results from simulated annealing. Both 2000Q and Advantage processors are able to compute returns that are consistently more than 80% of the optimal, except the two currency test cases where the algorithm fails to converge on the 2000Q.

**Figure 6 entropy-25-00541-f006:**
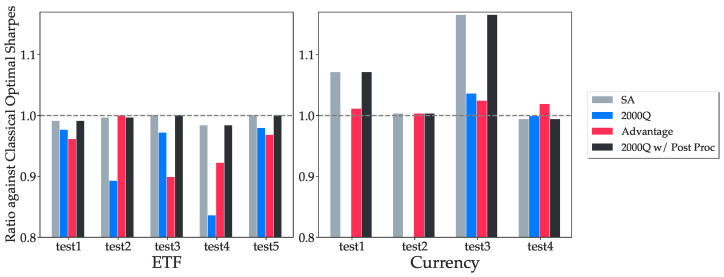
The comparison of the final Sharpe ratios between all four backends. Recall that the Sharpe ratio is the ratio of the return to the standard deviation of an asset for a set time period. Given a portfolio defined by the weight vector *w*, the Sharpe ratio of this portfolio is calculated as $\frac{\mu^{T}w}{\sqrt{w^{T}Cw}}$. A higher ratio means the backend can return portfolios with higher Sharpe ratios. The results confirm that the portfolio variances returned by the quantum processors are close to the optimal results obtained from classical optimization methods, and it is effective to solve standard constrained optimization problems as a QUBO.

**Figure 7 entropy-25-00541-f007:**
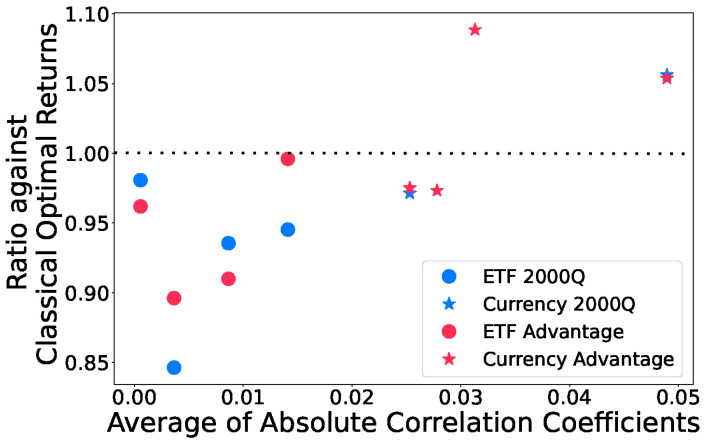
Final returns obtained from both quantum annealers against the average of the absolute correlation coefficients. The *x*-axis are the correlation coefficients of all *N* assets with each other computed using its daily returns from the chosen time periods and the *y*-axis is the ratio of the final returns against the classical optimal after Algorithm 1 converges using quantum annealers similar to Figure 5. The currency assets (stars) used in the tests all have higher correlation coefficients than those of the ETF assets, and generally yield better results.

**Figure 8 entropy-25-00541-f008:**
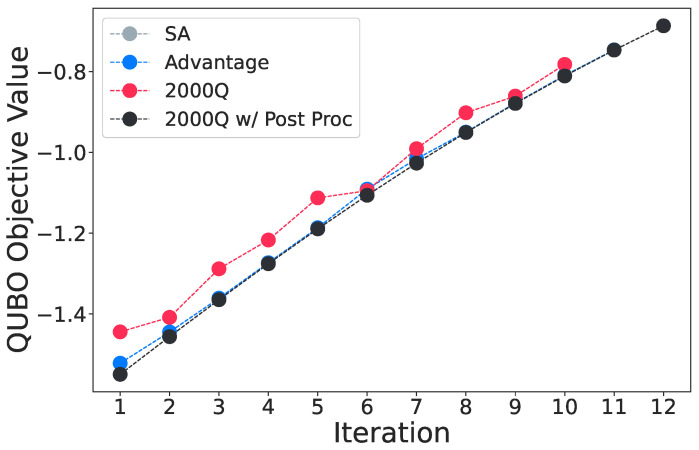
The objective comparison of the 12 asset test between all four backends. The solutions from 2000Q deviate from the ground states by a large margin, while the Advantage processor is able to keep up closely. Post-processing is able to improve the 2000Q results to once again match simulated annealing.

**Figure 9 entropy-25-00541-f009:**
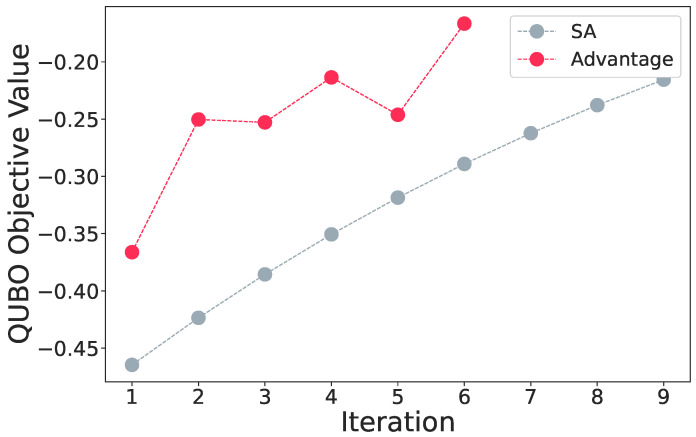
The objective comparison of the 23 asset test between simulated annealing and the Advantage processor. Due to the high chain lengths of the embedding, the Advantage processor fails to either reach the ground state or get close to it in all iterations, rendering the processor incapable of solving problems of such sizes.

**Table 1 entropy-25-00541-t001:** Embedding comparison on the 2000Q processor with 30 logical variables or 270 physical qubits after minor embedding. The objective value is calculated from Equation (Equation 18) and is normalized against the simulated annealer solving the same QUBO. All energies computed are negative, and their respective magnitudes are used for the comparison. The second embedding out of these four is able to find the solution with the better average and best objective value.

Embedding	Average Objective	Best Objective
1	95.66%	98.78%
2	96.83%	99.66%
3	96.53%	98.37%
4	96.21%	98.49%

**Table 2 entropy-25-00541-t002:** Embedding comparison on the Advantage processor with 30 logical variables or 134 physical qubits after minor embedding. Different embeddings on the Advantage processor show no statistically significant differences.

Embedding	Average Objective	Best Objective
1	99.25%	99.89%
2	99.52%	99.94%
3	99.22%	99.95%
4	99.48%	99.89%

**Table 3 entropy-25-00541-t003:** Objective values of last five iterations from simulated annealing and Advantage from the 12 asset test. This corroborates observations in Figure 8 that the Advantage processor is able to reach states with very good approximation ratios.

Last *k* Iteration	SA Objective	Advantage Objective	Difference
5	−1.026	−1.016	1.039%
4	−0.951	−0.950	0.092%
3	−0.879	−0.878	0.111%
2	−0.811	−0.809	0.170%
1	−0.746	−0.746	0.076%

## Data Availability

Data is available upon reasonable request.

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
