# Peer review of "Dynamic Asset Allocation with Expected Shortfall via Quantum Annealing"

_entropy, 2023, doi:10.3390/e25030541_

Round 1
Author Response
We thank the reviewer for the detailed comments. We agree we need to provide better arguments to support our choices of penalty coefficients in the QUBO. Additionally, we have improved the formatting across the board.
Here are our replies to the major comments and relevant changes in the paper have been marked as red:
- As the reviewer noted, there was no proof on whether the optimal solution to Eq.(14) was feasible for the Markowitz's optimization problem in Eq.(11) with our choices of penalty coefficients in line 229. However, we do not need to enforce this since our objective is to minimize the expected shortfall with a bilevel optimization approach. More specifically, we treat the constraints in Eq.(11) as soft constraints, i.e., small violations of their values are permitted. Portfolios from our QUBOs may not meet the target return exactly or have weights summing up to 1, but as long as the penalty is small, the solution is still of use to us.
- Both penalty terms in Eq.(14) are designed to have minimum values of -1. And to make sure the objective is not overwhelmed by them, we did scale it up to have a value around 1 (the scalar is anecdotal based on estimations of the objective values). These choices of penalty coefficients and objective scalars enabled us to construct QUBOs that lead to better portfolios in some of the tests in Section 4.4. However, we do have future research planned to gauge the effects of these parameters and find methods on how to pick them optimally.
Here are the responses to the minor comments:
- The variable in the algorithm has been changed to sigma_{ref_t} for consistency.
- The complexity notation in line 188 has been fixed.
- Eq.(13), previously Eq.(12), has been moved to one line.
- Eq.(14), previously Eq.(13), has been updated to have a similar format of Eq.(18).
- All references of equations have been updated to have the format of 'Eq. (x)'.
- Correct numbering has been added to the Previous Work subsection.
- Variable definitions for Eq.(19) have been updated to mention the subscript t, which is the time step.
- Eq.(20), previously Eq.(19), has been updated to the correct format of an optimization problem.
- Present tense is now used throughout the Previous Work section.
- We have updated the bibtex file. Most notably, arxiv manuscripts are now identified as [article] instead of [misc].
Reviewer 2 Report
The authors proposed a hybrid quantum-classical algorithm for solving the dynamic asset allocation problem and tested on the Dwave quantum annealers against some classical algorithms. The results look good and the whole paper is very well-written.
I only find a few typos which are listed below and definitely recommand it for publication in Entropy.
Corrections/typos:
1) Eq. (4) should have minus sign.
2) Line 180, O2^mn should be O(2^mn).
3) Eq. (8), H_B should be H_I?
4) Ref. [14], double titles.
5) Ref. [34], extra texts https://doi.org.
A comment:
From table 1 and 2, the quantum annealer never finds the true ground state of a 30 logical-variable problem. According to my own experiences, DWave quantum annealer, especially the advantage device, is able to find the true ground state for random fully-connected spin glass problem for small system size, for example, up to around 60 logical variables. It might be possible to tune the parameters for your case which enable the annealer to find the true ground state, at least for small problems.
Author Response
We thank the review for the recommendation and comments. We have updated the paper accordingly and here is the point-by-point response:
- Eq.(5), which was previously Eq.(4) now has the correct minus sign along with Eq.(6) and (7).
- The typo has been fixed.
- H_B is changed to H_I in Eq.(9), which was previously Eq.(8)
- Citation formats have been updated.
- Citation formats have been updated.
We agree quantum annealers can do better for the problems we have with optimized parameters. One particular area we are looking at for future work is to find ways to calibrate the hardware (shimming) for better performance.
Reviewer 3 Report
The contribution of this paper is the study of an important portfolio optimization problem on D-Wave.
The organization of the paper is very good and this makes it possible to follow their rationale and their conclusions. The equations, the Table and the Figure facilitate the understanding of the paper. I must admit that I found especially impressive and illuminating the Figures of Section 4; I commend the authors for them.
Having used D-Wave with my colleagues, I appreciated the analysis of Section 4 regarding the important question of optimally embedding the problem onto the existing topology. Ideally, I would like to know more about their conclusions regarding the violation of the soft constraint, as I have also found constraint violation to be an issue when programming on D-Wave. Of course, I understand that this is impossible, in view of the current state of the paper.
To cut a long story short, this paper is very well written and organized. The authors know this field very well. Furthermore, their analysis and results are interesting. Therefore, I believe that it can be published as is.
Round 2
Reviewer 1 Report
Now, after seeing the revision, and making a second detailed reading of the improved presentation of this manuscript, it is clear that the authors have no scientific methodology, scientific justification to choose their penalty parameters, nor any warrantee for the quality of the generated infeasible solutions.
Further, the original Markwitz model is a linearly constrained convex continuous optimization problem. That problem is solvable exactly in polynomial time, e.g., by using interior point methods.
It is not natural, and scientifically cannot be justified to approximate that efficiently solvable problem with binary expansion and unspecified penalty functions to NP-hard nonconvex discrete optimization problems.
This is a completely wrong approach to solve convex continuous portfolio optimization problems, thus this manuscript need to be rejected.